# Immunomodulation of Natural Killer Cell Function by Ribavirin Involves TYK-2 Activation and Subsequent Increased IFN-γ Secretion in the Context of In Vitro Hepatitis E Virus Infection

**DOI:** 10.3390/cells12030453

**Published:** 2023-01-31

**Authors:** Paul Kupke, Akinbami Adenugba, Mathias Schemmerer, Florian Bitterer, Hans J. Schlitt, Edward K. Geissler, Jürgen J. Wenzel, Jens M. Werner

**Affiliations:** 1Department of Surgery, University Hospital Regensburg, 93053 Regensburg, Germany; 2National Consultant Laboratory for HAV and HEV, Institute of Clinical Microbiology and Hygiene, University Hospital Regensburg, 93053 Regensburg, Germany

**Keywords:** ribavirin, hepatitis E virus, transplantation, immunosuppression, NK cells

## Abstract

Hepatitis E virus (HEV) is a major cause of acute hepatitis globally. Chronic and fulminant courses are observed especially in immunocompromised transplant recipients since administration of ribavirin (RBV) does not always lead to a sustained virologic response. By in vitro stimulation of NK cells through hepatoma cell lines inoculated with a full-length HEV and treatment with RBV, we analyzed the viral replication and cell response to further elucidate the mechanism of action of RBV on immune cells, especially NK cells, in the context of HEV infection. Co-culture of HEV-infected hepatoma cells with PBMCs and treatment with RBV both resulted in a decrease in viral replication, which in combination showed an additive effect. An analysis of NK cell functions after stimulation revealed evidence of reduced cytotoxicity by decreased TRAIL and CD107a degranulation. Simultaneously, IFN-ɣ production was significantly increased through the IL-12R pathway. Although there was no direct effect on the IL-12R subunits, downstream events starting with TYK-2 and subsequently pSTAT4 were upregulated. In conclusion, we showed that RBV has an immunomodulatory effect on the IL-12R pathway of NK cells via TYK-2. This subsequently leads to an enhanced IFN-ɣ response and thus, to an additive antiviral effect in the context of an in vitro HEV infection.

## 1. Introduction

The *Paslahepevirus balayani* (hepatitis E virus, HEV) is a leading cause of acute hepatitis, globally. In the majority of cases, the infection progresses asymptomatically and is self-limiting. However, severe courses occur in certain patient groups [1,2,3]. Genotype 3 is mainly transmitted zoonotically through undercooked game and wild boar products in industrialized countries. This plays a crucial role in immunocompromised patients as the infection can become chronic and end fatally [4,5]. In fact, most patients with persistent HEV infection are solid organ transplant (SOT) recipients [6].

Currently, there are very few treatment options for complicated HEV infections. At first, in patients after SOT, immunosuppression is reduced carefully, always with the substantial risk of organ rejection. This has been successful in about one-third of patients [7,8]. Next treatment option is ribavirin (RBV). It is a nucleoside analogue, which has been used mainly in the treatment of *Hepacivirus C* (hepatitis C virus, HCV) infections [9]. A first course of RBV therapy leads to a sustained virologic response (SVR) in up to 81% of HEV patients with a persistent infection [8,10,11]. The exact mechanisms of RBV on HEV clearance remain unclear. It is certain that it acts both directly on viral replication and as an immunomodulatory agent [10]. Unfortunately, side effects, which are often not negligible, play a dominant role and result in the necessity to discontinue treatment [10]. 

Most recently, the focus shifted towards sofosbuvir, an agent that is also known to be used in HCV therapy [12]. It has been shown in in vitro experiments that it decreases HEV replication and enhances the effect of RBV [13]. Furthermore, the additional administration of sofosbuvir in patients with ongoing RBV therapy led to an additive effect or even a temporary eradication of the virus [14,15,16]. However, monotherapy with sofosbuvir did not show a decisive effect on the elimination of the virus although replication was reduced [17]. 

In general, it is still largely unclear which patients respond to the different treatment options. This is primarily due to the fact that the complex interplay of the immune responses of the different cell populations after treatment is not yet fully understood. In this context, evidence can be provided by studies on the immune response to other viral hepatitis infections. Natural killer (NK) cells in particular exert a significant early impact on viral hepatitis pathogens mainly through direct cytotoxicity and the production of antiviral substances such as interferon-γ (IFN-γ) [18,19]. They are defined as CD3^−^ CD56^+^ lymphoid cells [20]. Inhibitory signals dominate NK cells in the steady state. In response to pathogens, NK cells become activated either by decreased signals from inhibitory receptors or by increased signals from activating receptors [18,19]. In addition, pro-inflammatory cytokines, such as interleukin (IL)-12, IL-15, or IL-18, play an important role [21]. It has been shown that in patients with an HEV infection, NK cells are more activated and show an elevated fraction of CD56^bright^ primarily cytokine-producing NK cells. Furthermore, there is evidence of migration from the periphery to the liver, the site of infection [22,23]. Liver biopsies revealed that especially in severe cases, HEV-infected patients show the highest numbers of intrahepatic NK cells compared to other viral hepatitis [22]. Previous studies have shown that RBV treatment in the setting of HCV infection results in an enhanced NK cell IFN-γ response [24,25]. However, the exact mechanism remains unclear.

In this study, we present results on the immunomodulation of RBV on NK cells in the context of an in vitro HEV infection and propose a mechanism by which RBV increases IFN-γ production by NK cells.

## 2. Materials and Methods

### 2.1. HEV Culture System

The human hepatoma cell line, HepaRG, was obtained from Biopredic International, Saint-Grégoire, France. The cells were raised in 12 well-plates (TPP, Trasadingen, Switzerland) or in T75 flasks (TPP) at 2.5 × 10^4^ cells/cm^2^ in HepaRG growth medium (William’s E medium, Fisher Scientific, Waltham, MA, USA) supplemented with 10% HyClone FetalClone II (Fisher Scientific), 1% penicillin/streptomycin (Sigma-Aldrich, St. Louis, MO, USA), 1% L-glutamine (Sigma-Aldrich), 0.023 IE/mL insulin (Sanofi, Paris, France), 4.7 µg/mL hydrocortisone (Pfizer, New York, NY, USA), and 80 µg/mL gentamicin (Ratiopharm, Ulm, Germany) at 37 °C and 5% CO_2_ for two weeks and for an additional two weeks, 1.8% dimethyl sulfoxide (DMSO, AppliChem, Darmstadt, Germany) was added to aid the differentiation into hepatocyte-like cells. The medium was changed 2 times per week. After four weeks of culture, the HepaRG cells were inoculated with approximately 10^8^ copies/mL (c/mL) of HEV subtype 3c strain 14-16753 [26] (GenBank accession number MK089849) and the cells were maintained for another week at 34.5 °C and 5% CO_2_. To generate the HEV supernatants for inoculation, infected hepatoma cells were cultured, and the supernatants were obtained periodically and stored at −80 °C until further use in experiments.

### 2.2. PBMCs Co-Culture

PBMCs were isolated by density gradient centrifugation from healthy, HEV-negative-tested platelet donors as a byproduct of apheresis. They were then added at an effector to target ratio of 1:1. Unlabeled CD56^+^ or CD19^+^ cells were isolated or depleted with the NK Isolation Kit (Miltenyi Biotec, Bergisch Gladbach, Germany) or the corresponding CD56 or CD19 MicroBeads (Miltenyi Biotec) by autoMACS Pro Separator (Miltenyi Biotec) according to the manufacturer’s protocol. PBMCs were stored in RPMI 1640 medium (Fisher Scientific) containing 10% DMSO at –160 °C in liquid nitrogen. For culturing, the cells were thawed in a 37 °C water bath and the suspension was then transferred in Dulbecco’s phosphate buffered saline (DPBS, PAN-Biotech, Aidenbach, Germany) containing 100 µg/mL DNase I (AppliChem) followed by slow dilution. The co-culture was carried out with a medium containing RPMI 1640 + GlutaMAX (Fisher Scientific), 10% HyClone FetalClone II serum, 1% sodium pyruvate (Fisher Scientific), 1% MEM nonessential amino acids (Fisher Scientific), 7.5% sodium bicarbonate (Fisher Scientific), 1% penicillin/streptomycin, and 50 mM 2-mercaptoethanol (Sigma-Aldrich). RBV (Sigma-Aldrich) and sofosbuvir (SOF, AmBeed, Arlington Heights, IL, USA) treatments were performed in a concentration of 500 µM and 100 µM, respectively. The duration of the co-culture was 24 h. Inhibition with cerdulatinib (CER, Selleck Chemicals, Houston, TX, USA) was performed at 1 nM for 48 h prior to RBV treatment.

### 2.3. HEV RNA Quantification

Viral RNA was isolated from the HepaRG cell pellet using TRIzol (Invitrogen, Waltham, MA, USA) according to the manufacturer’s protocol. The RT-PCR was then performed by a LightCycler 480 (Roche, Basel, Switzerland) with NxtScript RT Reaction Mix and Enzyme Solution (Roche), a JHEV MGB probe (sequence TGA TTC TCA GCC CTT CGC, Applied Biosystems, Waltham, MA, USA), and forward (sequence GGT GGT TTC TGG GGT GAC) and reverse (sequence AGG GGT TGG TTG GAT GAA) primers (Metabion, Planegg-Steinkirchen, Germany) [27,28].

### 2.4. HEV Antigen ELISA

To analyze the HEV antigen, HEV-Ag ELISA (Wantai, Beijing, China) was performed on the supernatants collected from the culture according to the manufacturer’s protocol.

### 2.5. Luciferase Assay

HepaRG cells were transduced with EF1a-luciferase (firefly)-2A-GFP in a lentiviral transduction (Amsbio, Abingdon, UK) according to the manufacturer’s recommendations. After cell expansion under selection pressure using blasticidin (InvivoGen, San Diego, CA, USA) with a concentration of 1.5 µg/mL and multiple flow cytometric sorting, a luciferase^+^ HepaRG culture with a purity of >95% could be generated.

To determine the extent of the PBMC-mediated killing of HepaRG cells, the cell supernatant was removed, the cells were washed with DPBS, and the cells were lysed with 40 µL 10% Triton-X buffer for 15 min at room temperature. Immediately after the addition of 60 µL of luciferin buffer containing 0.5 mM D-luciferin (Biosynth, Staad, Switzerland) with AMP and ATP, the emitted light was measured at 560 nm with a VarioSkan (ThermoFischer).

### 2.6. Stimulation Prior to Staining

Where stated, cells were stimulated with combinations of recombinant human IL-12 (R&D Systems, Minneapolis, MN, USA), IL-15 (R&D Systems), and IL-18 (R&D Systems) for 12 h at concentrations of 0.5 ng/mL, 20 ng/mL, and 100 ng/mL [24,25,29,30], respectively, and protein transport inhibitors containing brefeldin A (BD Biosciences, Franklin Lakes, WI, USA) and monensin (BD Biosciences) were added for the last 4 h. For analyzing CD107a degranulation, cells were stimulated with K562 cells (ATCC, Manassas, VA, USA) at an effector to target ratio of 1:1 and anti-CD107a was added during the last 6 h.

### 2.7. Staining for Flow Cytometry

At first, dead cells were excluded by adding ethidium monoazide bromide (Sigma-Aldrich) for 10 min on ice under direct light. Then, the cells were stained for CD14, CD19, and CD3 to exclude monocytes, B cells, and T cells, respectively. The antibodies for CD56 and CD16 were used to identify the NK cells. Bright NK cells were defined as CD56^++^ and CD16^−^. The gating strategy is shown in Figure 2A. Regular staining was carried out for 30 min at 4 °C and phosflow staining for 25 min at room temperature. For intracellular staining, cells were fixed and permeabilized with the Cytofix/Cytoperm kit (BD Biosciences) and for phosflow staining with the Cytofix and Phosflow Perm Buffer III kit (BD Biosciences). The FACSCanto II flow cytometer (BD Biosciences) and FACSDiva software (BD Biosciences) were used to obtain the data, and the FlowJo v10 (BD Biosciences) was used to analyze the data.

The following antibodies for surface staining were used: anti-CD14-PerCP-Cy5.5 (BD Biosciences), anti-CD19-PerCP-Cy5.5 (BD Biosciences), anti-CD3-APC-Cy7 (BioLegend, San Diego, CA, USA), anti-CD56-PE-Cy7 (BD Biosciences), anti-CD16-Pacific Blue (BD Biosciences), anti-CD38-PE (BD Biosciences), anti-NKp44-PE (Beckman Coulter, Brea, CA, USA), anti-NKp46-APC (Miltenyi Biotec), anti-NKp80-FITC (Miltenyi Biotec), anti-NKG2A-APC (Beckman Coulter), anti-NKG2C-PE (Miltenyi Biotec), anti-NKG2D-PE (Beckman Coulter), anti-CD244-FITC (BioLegend), anti-TRAIL-PE (BD Biosciences), anti-CD107a-PE (BD Biosciences), anti-CD69-APC (BD Biosciences), anti-IL-12Rβ1-APC (BD Biosciences), and anti-IL-12Rβ2-Alexa Fluor 488 (R&D Systems). The following antibodies were used for intracellular staining: anti-IFN-γ-Alexa Fluor 488 (BioLegend), anti-TYK-2-Alexa Fluor 647 (Santa Cruz Biotechnology, Dallas, TX, USA), and anti-JAK-2-FITC (Santa Cruz Biotechnology). The antibodies used in the phosflow protocol were the following: anti-CD20-PerCP-Cy5.5 (BD Biosciences), anti-CD56-PE (Beckman Coulter), anti-CD3-FITC (BD Biosciences), anti-pSTAT1-VioBlue (Miltenyi Biotec), anti-pSTAT3-Pacific Blue (BD Biosciences), and anti-pSTAT4-Alexa Fluor 647 (BD Biosciences).

### 2.8. PCR for TYK-2 and IFN-γ Expression

All steps were performed according to the manufacturer’s protocol. RNA was isolated from the cell pellet using TRIzol. The RNA was then reverse-transcribed by using the AffinityScript qPCR cDNA synthesis kit (Agilent Technologies, Santa Clara, CA, USA). PCR was thereafter performed using the LightCycler 480 SYBR Green I reaction mix (Roche) and GAPDH as the housekeeping gene. The following primers (Metabion) were used: GAPDH forward (CGA CCA CTT TGT CAA GCT CA), GAPDH reverse (AGG GGA GAT TCA GTG TGG TG) [31], TYK-2 forward (AGC CAT CTT GGA AGA CAG CAA), TYK-2 reverse (GAC TTT GTG TGC GAT GTG GAT) [32], IFN-γ forward (TGA CCA GAG CAT CCA AAA GA), and IFN-γ reverse (CTC TTC GAC CTC GAA ACA GC) [31].

### 2.9. Statistics

Statistical analyses were performed using the GraphPad Prism v9 (GraphPad Software, Boston, MA, USA). Tests were used as indicated in the respective figure legends. The *p*-values of the statistical tests were calculated as two-tailed tests where applicable. Only *p*-values of < 0.05 were considered significant.

## 3. Results

### 3.1. RBV Exerts an Additive Effect with PBMCs on HEV Replication, Which Is Driven by NK Cells to a Substantial Amount

It has proven to be rather difficult to establish valid cell culture models that realistically simulate an HEV infection in vitro. Frequently, existing models show weak viral replication or limited reproducibility, which significantly diminishes their practical value and validity [5]. We were able to reliably infect human hepatoma cell lines with a well-characterized HEV subtype 3c strain and demonstrate stable replication, which served as the basis for our experiments [26].

Inoculation of HepaRG cells with HEV-3c resulted in a constantly high replication in the cell pellet (median concentration 7.6 × 10^6^ c/mL, conversion factor 1.00 IU/mL = 1.15 c/mL), which was attenuated after the co-culture with PBMCs (4.6 × 10^6^ c/mL, *p* = 0.0048). The same effect occurred when HEV^+^ HepaRG cells were treated with RBV (4.5 × 10^6^ c/mL, *p* < 0.0001). Interestingly, the combined administration of RBV and PBMCs led to an additive effect that further decreased viral replication (2.6 × 10^6^ c/mL, *p* = 0.0009), as shown in Figure 1A.

To rule out a direct effect of PBMCs on healthy HepaRG cells and to validate our co-culture system, we inoculated luciferase-tagged HepaRG cells and analyzed the effects of the PBMC co-culture on luciferase activity. As shown in Figure 1B, these results demonstrate that the relative light units (RLU) were primarily reduced in the co-culture with the inoculated HEV^+^ HepaRG cells when compared to the untreated HEV^−^ controls (fold change 0.905 vs. 0.932, *p* = 0.0078).

By determining antigen titers from the supernatants of our co-cultures (Figure 1C), it was possible to detect a significant release of HEV antigen. While this was evident when the PBMCs were added alone (fold change 1.471, *p* < 0.0001), treatment with RBV did not result in any significant increase in titers (fold change 1.101, *ns*). In agreement with our PCR experiments, the combined administration of PBMCs and RBV revealed an additive effect, which substantially increased the release of HEV antigen (fold change 2.271, *p* < 0.0001).

As NK cells play a substantial role in the clearance of viral pathogens, including HEV, as described on several occasions [19,22,23], we investigated the effect of HEV on the IL-12-/IL-15-stimulated IFN-γ production of NK cells in Figure 1D,E. Here, a marked increase was evident after contact with the infected target cells (MFI 1297 vs. 1859, *p* = 0.0312), which was further elevated by the additional treatment with RBV (MFI 923 vs. 1339, *p* = 0.0479). Moreover, as shown in Figure 1F, the depletion of CD56^+^ cells resulted in a diminished reduction of viral titers compared to the treatment with undepleted PBMCs, which was as expected not apparent after the depletion of CD19^+^ cells (1.29 vs. 0.93; *p* = 0.0425).

In summary, our experiments demonstrate that the addition of PBMCs to HEV-inoculated HepaRG cells decreased viral replication. Thereby, the treatment with RBV revealed a similar effect. Simultaneous administration of PBMCs and RBV unmasked an additive effect that significantly reduced viral titers and released an augmented amount of HEV antigen. Moreover, this effect was dependent on the NK cells.

### 3.2. Divergent Effect of RBV Treatment on NK Cell Receptors

Following the phenotyping of NK cells after RBV treatment as demonstrated in Figure 2, an increased NK cell activation could be detected by means of the surface molecule CD38 (mean fluorescence intensity (MFI) 3439 vs. 3837, *p* < 0.0001), which is primarily regarded as an indicator of activation [33]. The effect on activatory and inhibitory receptors, however, was divergent [19,34]. The natural cytotoxic receptor NKp44 was decreased by RBV (MFI 348 vs. 311, *p* < 0.0001) whereas NKp46 and NKp80 were increased (MFI 3309 vs. 3501, *p* = 0.0139 and MFI 1070 vs. 1109, *p* = 0.0005, respectively). In addition, a differing reaction of the C-type lectin receptors was observed. NKG2C was increased (MFI 269 vs. 290, *p* = 0.0332) while NKG2D was strongly decreased (MFI 1156 vs. 839, *p* < 0.0001). At the same time, there was a clear decrease of CD244 (MFI 1853 vs. 1685, *p* = 0.0005), a surface protein to which stimulatory and inhibitory functions are attributed [35,36]. Furthermore, the inhibitory receptor NKG2A showed a decreased expression after RBV treatment (MFI 2980 vs. 2742, *p* = 0.0022). Corresponding frequencies are demonstrated in Appendix A.

### 3.3. RBV Treatment Diminishes Cell-Mediated Cellular Cytotoxicity, but Enhances IFN-γ Production upon IL Stimulation, Which Is Mainly Driven through IL-12

NK cells exert cytotoxic effects via two main mechanisms. On the one hand, through the release of granules containing perforin and granzymes and on the other, through the induction of apoptosis via the expression of TRAIL on their surface [37].

As demonstrated in Figure 3A, TRAIL expression was reduced after RBV treatment, both in the total NK cell population (MFI 472 vs. 415, *p* = 0.0087 and 10.3% vs. 6.1%, *p* = 0.0139) and especially among bright NK cells (MFI 729 vs. 585, *p* = 0.0003 and 34.8% vs. 15.0%, *p* = 0.0004). Simultaneously, after stimulation with K562 cells, a cell line without expression of the MHC complexes, there was less CD107a degranulation by the NK cells (MFI 258 vs. 212, *p* = 0.0002 and 6.8% vs. 5.4%, *p* < 0.0001) and by the dim NK cells subpopulation (MFI 259 vs. 212, *p* = 0.0003 and 6.9% vs. 6.0%, *p* < 0.0001), as well (Figure 3B).

To investigate the activity and the ability of NK cells to produce IFN-γ, the cells were stimulated with IL-12 and IL-15 (Figure 3C,D). The results revealed that both the total population and the bright population, which is predominantly responsible for IFN-γ production, showed an increase in CD69 expression (MFI 12379 vs. 14431, *p* < 0.0001 and 58.4% vs. 64.1%, *p* < 0.0001 and MFI 16976 vs. 20549, *p* < 0.0001 and 78.1% vs. 84.7%, *p* < 0.0001, respectively) and IFN-γ production (MFI 1020 vs. 1394, *p* < 0.0001 and 24.5% vs. 25.9%, *p* < 0.0001 and MFI 3450 vs. 4556, *p* = 0.0032 and 55.3% vs. 65.7%, *p* < 0.0001, respectively). This effect on IFN-γ production was also demonstrated in isolated NK cells, both for total NK cells (MFI 736 vs. 790, *p* < 0.0001 and 24.8% vs. 26.1%, *p* = 0.0066) and bright NK cells (MFI 1473 vs. 1851, *p* = 0.0013 and 48.2% vs. 49.9%, *p* = 0.0002).

Of note, the antiviral agent, sofosbuvir (SOF), which is currently attracting some interest in HEV therapy, showed no comparable immunomodulatory effects in our study (see Appendix A).

### 3.4. Increased IFN-γ Production by RBV-Treated NK Cells Is Primarily Achieved via Upregulation of TYK-2 as Part of the IL-12 Receptor Pathway

To further elucidate how RBV subsequently promotes IFN-γ production by NK cells, different common in vitro IL stimulations [19] were investigated in combination with RBV treatment (Figure 4). Stimulation with IL-12 alone (Change MFI 46, *p* = 0.0127 and 1.3%, *p* = 0.0123) and especially in combination with IL-15 (Change MFI 415, *p* = 0.0082 and 3.6%, *p* = 0.0001) led to a strong induction of IFN-γ production in RBV-treated NK cells. This effect was in particular evident in bright NK cells (Change MFI 95, *p* = 0.0041 and 2.2%, *p* < 0.0001 and Change MFI 2538, *p* < 0.0001 and 13.5%, *p* < 0.0001, respectively).

Interestingly, as shown in Figure 5A, RBV-treated NK cells showed a slight downregulation of both IL-12 receptor subunits, IL-12Rβ1 (MFI 481 vs. 433, *p* = 0.0005 and 79.3% vs. 73.7%, *p* = 0.0005) and IL-12Rβ2 (MFI 104 vs. 101, *p* = 0.0078 and 33.9% vs. 32.3%, *p* = 0.0269). At the same time, the Janus kinases, JAK-2 (see Appendix A) and especially TYK-2 (total: MFI 421 vs. 476, *p* < 0.0001 and 34.2% vs. 42.4%, *p* = 0.0025 and bright: MFI 518 vs. 562, *p* < 0.0001 and 57.6% vs. 64.8%, *p* = 0.0461), showed an increase upon IL-12 and IL-15 stimulation due to RBV treatment (Figure 5B). Selective TYK-2 inhibition with cerdulatinib led to an attenuated IFN-γ response after RBV treatment, which was most evident in bright NK cells, the predominant IFN-ɣ-producing subpopulation (See Appendix A).

Subsequently, as shown in Figure 5C, phosphorylation of STAT4 was increased (total: MFI 450 vs. 478, *p* = 0.0004 and 66.3% vs. 70.4%, *p* = 0.0023 and bright: MFI 452 vs. 497, *p* < 0.0001 and 74.0% vs. 77.8%, *p* = 0.0005) whereas pSTAT1 and pSTAT3 showed no increase according to our results (see Appendix A). As shown in Figure 5D for different timepoints of IL-12/IL-15 stimulation after RBV treatment of NK cells, an increase of TYK-2 was confirmed on the transcriptomic level by RT-PCR after 2 h (2^−ΔΔCt^ 1.329, *p* = 0.0340) and a subsequent increase of IFN-γ (2^−ΔΔCt^ 1.653, *p* = 0.0020) after 12 h.

Taken together, RBV, starting with the upregulation of TYK-2, interferes with the IL-12 cascade and ultimately leads to an increased IFN-γ production (Figure 6), which has shown to be crucial in the immune response to viral hepatitis.

## 4. Discussion

Our study investigated the immunomodulation of ribavirin (RBV), a virostatic drug that has gained prominence in *Hepacivirus C* (hepatitis C virus, HCV) therapy and has established itself as an important component in the therapy of complicated *Paslahepevirus balayani* (hepatitis E virus, HEV) infections [8]. Due to its wide spectrum of side effects, it is not suitable for some patient groups, but it is an important off-label treatment option for HEV, especially for patients after solid organ transplant (SOT) in whom the reduction of immunosuppression alone has failed [10]. After the direct antiviral effect was in the spotlight at the beginning [38], it emerged that RBV also acts in an immunomodulatory way [39]. Especially regarding NK cells, the number of studies in the context of HEV infections is very scarce. Our study shows how RBV, in addition to its direct antiviral effect, influences the immune response of NK cells in a divergent manner and in addition to a reduced direct cytotoxicity, promotes increased IFN-γ production.

In the investigation of HEV infections, it is rather difficult to acquire appropriate patient samples [5]. On one hand, it is often not possible to identify patients due to the mostly asymptomatic acute cases. On the other, the existing animal and cell culture models show fundamental limitations [40,41]. Nevertheless, based on the work of Schemmerer et al. [26], we were able to establish a robust, co-cultivatable cell culture model based on HepaRG persistently infected with an HEV subtype 3c strain yielding high viral replication.

In clinical practice in industrialized countries, RBV therapy primarily plays a role in the off-label treatment of chronic HEV infections in SOT recipients [8]. According to the study by Kamar et al. [10], initially 95% of all transplant patients with persistent HEV infection showed an adequate response to therapy with RBV. However, SVR after a first course of RBV is still not achieved in 19% of patients, which can be decreased to 10% with a second period of treatment [11]. The only predictive factor in these patients was a lower lymphocyte count at the start of therapy. It has been shown that different subtypes such as HEV-3efg cause more severe courses compared to other HEV-3 subtypes [42], but there is disagreement about whether different HEV RNA variants have an impact on the outcome of RBV therapy [11,43]. Debing et al. [44], for instance, showed an association of the G1634R mutation of the HEV polymerase with the failure of RBV therapy. Another study gave evidence for the development of this mutation during RBV therapy [45]. However, the detection of the mutation before the start of therapy did not indicate RBV resistance in the end [46]. A possible further escalation of therapy in the case of nonresponse to RBV is pegylated interferon alfa, which, however, should only be used after careful consideration due to its manifold side-effects and the high risk of graft rejection [5]. Thus, there are limited alternatives to sofosbuvir, which recently failed in its hoped-for benefit as a monotherapeutic agent [17]. It is therefore necessary to further optimize the therapy with RBV and to break down its immunomodulatory effect in further detail, especially to identify and cover those patients in an early stage whose treatment does not result in SVR.

Interestingly, phenotyping of NK cells showed divergent findings. While RBV led to an increased expression of activatory receptors and decreased expression of inhibitory receptors, there was a marked downregulation of the activatory receptors NKp44, NKG2D, and CD244. Nevertheless, these results are not surprising since all three play a major role in the direct cytotoxicity of NK cells [47,48,49], which as shown by our data, was markedly reduced.

Multiple viruses such as the *Hepatitis B virus* are known to promote the induction of IL-12 [50]. Our study demonstrates that RBV acts as an IL-12 sensitizer and increases the sensitivity of NK cells to IL-12 stimulation. It has been shown in the context of HCV infections that RBV leads in vivo to a differentiation towards a type 1 T-cell response by increased amounts of IL-12 [51] or by upregulation of the IL-12 receptor [52]. This is partly consistent with our in vitro findings on NK cells, which, however, do not indicate an approach at the receptor itself, but in the following intracellular cascade starting with TYK-2. In fact, the subunits of the receptor even showed a slight downregulation after RBV stimulation, presumably following a negative feedback mechanism. Furthermore, also in the setting of HCV therapies, it has been shown that RBV leads to an improved IFN-γ response through increased phosphorylation of STAT4 [24] as well as it sensitizes the cell to IFN-based therapy regimens in this regard [25]. Interestingly, Markova et al. [53] did not detect any changes by RBV in NK cell phenotype nor function in their HCV studies both in vivo and in vitro. Ogbomo et al. [54] even demonstrated a mechanism by which RBV decreases each direct cytotoxicity and IFN-γ production via the IL-15 receptor, which is merely consistent with our observations of decreased direct cytotoxicity before and after IL-12 stimulation.

TYK-2 as a therapeutic target plays its role mainly in the context of nonspecific inhibition of Janus kinases in the therapy of autoimmune and inflammatory diseases [55,56]. Recently, due to safety concerns, interest has shifted towards the specific inhibition of Janus kinases. Direct inhibition of TYK-2 is currently being discussed primarily in the therapy of dermatological autoimmune diseases where its efficacy and safety are being tested in clinical trials [57].

The primary limitation of our study is that our findings were obtained using healthy donor lymphocytes in the setting of an HEV co-culture system. To validate these in vitro findings, studies on clinical patient samples are urgently needed and in the best case, also by patients in whom therapy with RBV failed to reach SVR in the end.

In conclusion, we identified an immunomodulatory mechanism of RBV on NK cells by which, starting with TYK-2 via the subsequent transcription factor cascade through elevated phosphorylation of STAT4, it increases the sensitivity of NK cells to stimulation with IL-12 and subsequently leads to a strongly increased IFN-γ response in the context of an HEV infection.

## Figures and Tables

**Figure 1 cells-12-00453-f001:**
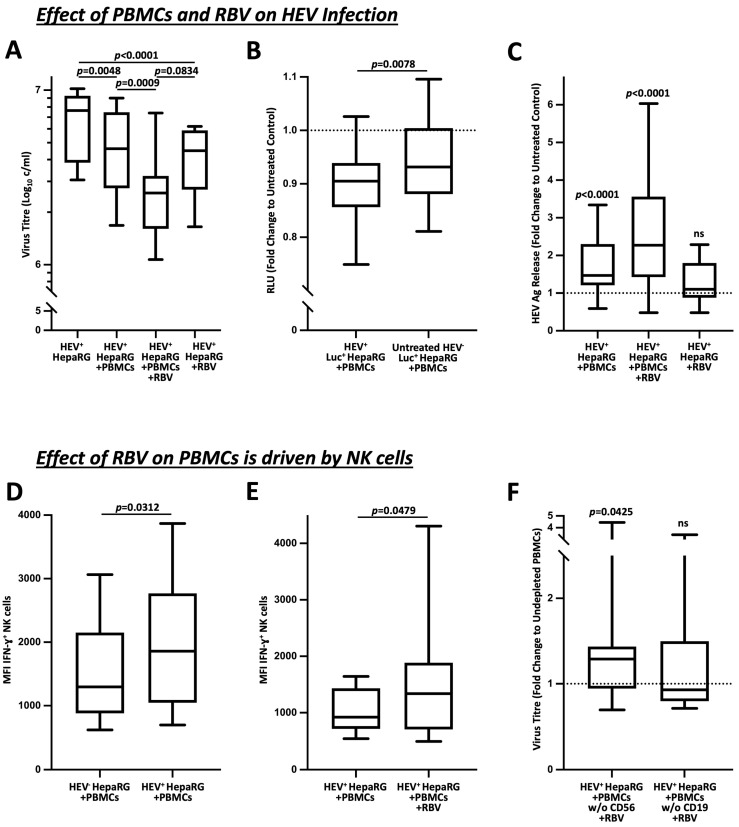
Immunomodulation of RBV boosts the antiviral effect of NK cells. (**A**) demonstrates the effect of RBV on PBMCs in HEV^+^ HepaRG co-culture represented by alteration in viral loads (n = 24). (**B**) shows the change of luciferase activity as indication for HepaRG viability in sole presence of PBMCs. The corresponding setups without co-culture of PBMCs acted as the untreated controls (n = 39). (**C**) pictures the quantification of HEV antigen concentrations from co-culture supernatants in relation to untreated controls (n = 24). (**D**,**E**) show the effects of co-culture of PBMCs with infected target cells and RBV treatment on IFN-γ production by NK cells upon IL-12/IL-15 stimulation (n = 6 and n = 13). (**F**) illustrates the effect of CD56 depletion on antiviral activity by PBMCs in the context of RBV treatment (n = 26). Appearance: median with interquartile range (whiskers min to max), statistical analysis: Friedman test with Dunn’s multiple comparisons test (**A**), Wilcoxon matched-pairs signed-rank test (**B**,**D**,**E**) or one-sample Wilcoxon signed-rank test (**C**,**D**).

**Figure 2 cells-12-00453-f002:**
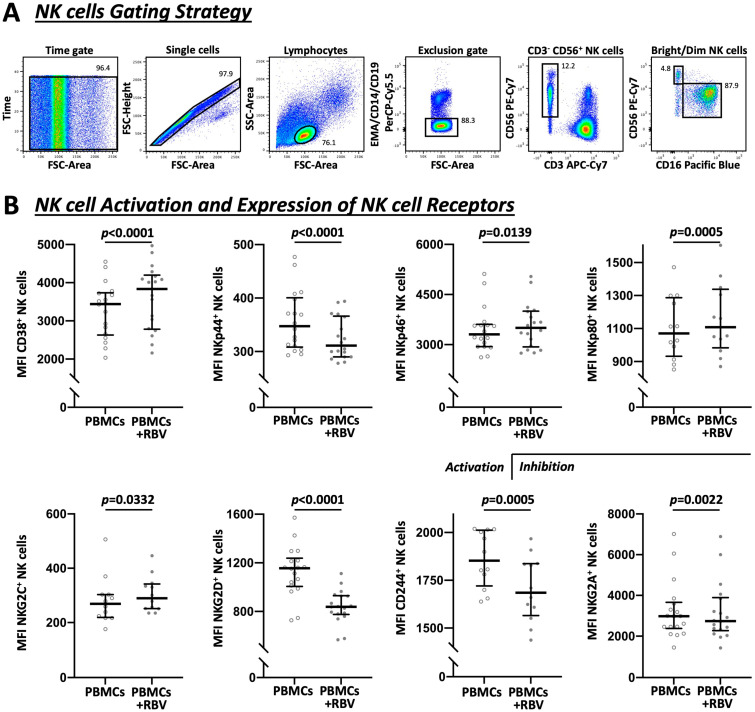
Flow cytometric analysis of NK cells and involved receptors under RBV treatment. (**A**) Exemplary gating strategy to show the sequence of identifying NK cells as CD3^−^CD56^+^ PBMCs as well as the fractionation in dim (CD56^+^CD16^+^) and bright (CD56^++^CD16^−^) subpopulations. (**B**) Expression of affected NK cell receptors shown by MFI (n = 18 and n = 12). Appearance: median with interquartile range, statistical analysis: Wilcoxon matched-pairs signed-rank test.

**Figure 3 cells-12-00453-f003:**
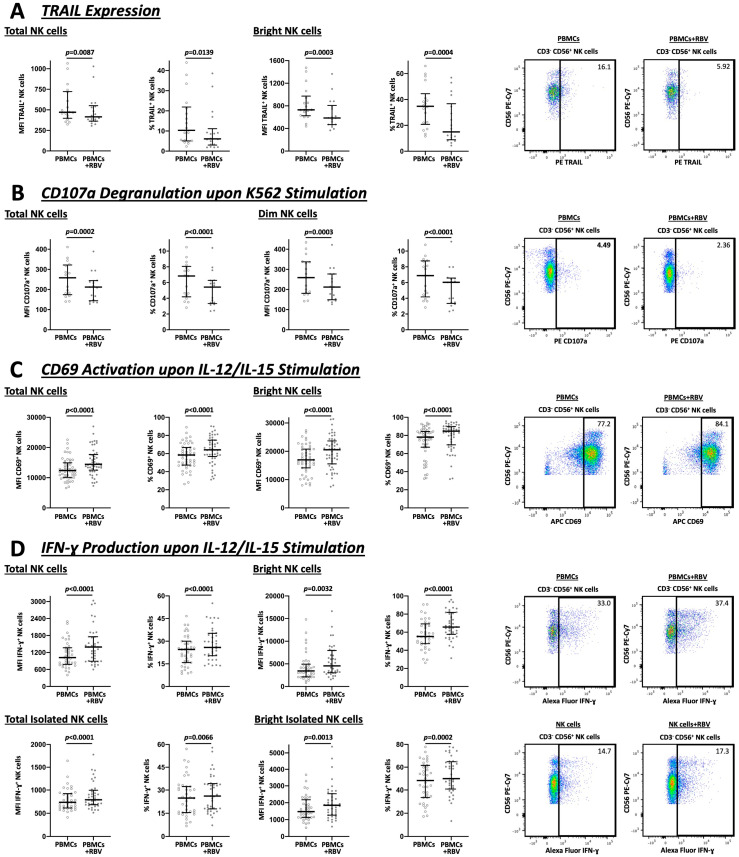
Flow cytometric analysis of total and bright or dim NK cells under RBV treatment with exemplary gating strategy. MFI and frequencies compared. (**A**) Expression of TRAIL by NK cells (n = 18). (**B**) NK cell degranulation analyzed by CD107a expression upon K562 stimulation (n = 17). (**C**) Activation of NK cells represented by CD69 expression upon IL-12/IL-15 stimulation (n = 51). (**D**) Analysis of IFN-γ upon IL-12/IL-15 stimulation as a major indicator for NK cell cytokine production (n = 39). Appearance: median with interquartile range, statistical analysis: Wilcoxon matched-pairs signed-rank test.

**Figure 4 cells-12-00453-f004:**
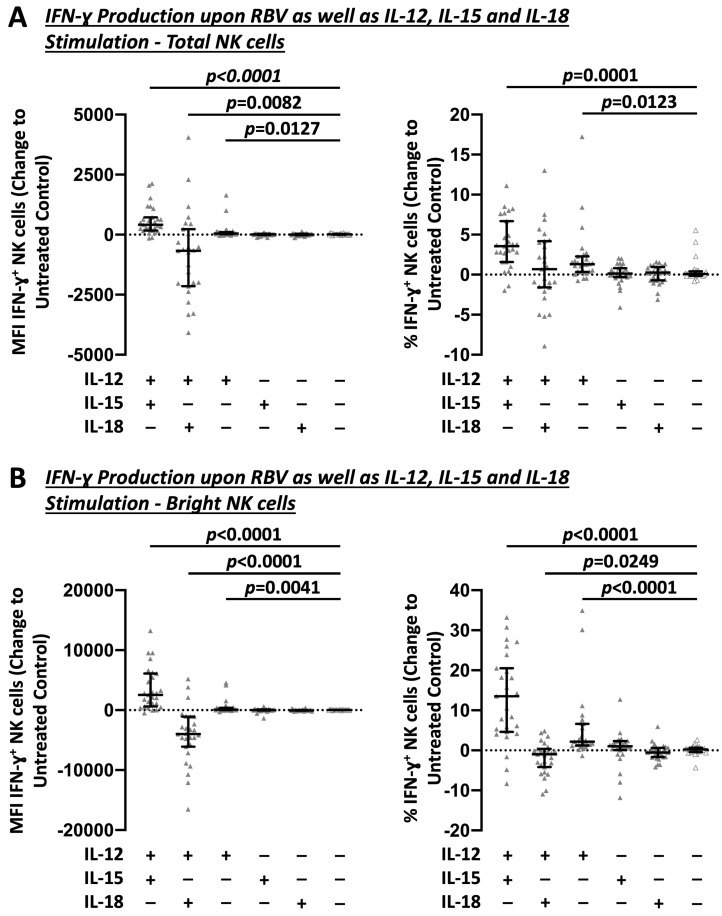
Impact of different sole and combined IL stimulations on RBV-treated NK cells. Flow cytometric analysis of IFN-γ production as change to untreated controls. MFI and frequencies compared. (**A**) Total NK cells. (**B**) Bright NK cells (n = 27). Appearance: median with interquartile range, statistical analysis: Wilcoxon matched-pairs signed-rank test.

**Figure 5 cells-12-00453-f005:**
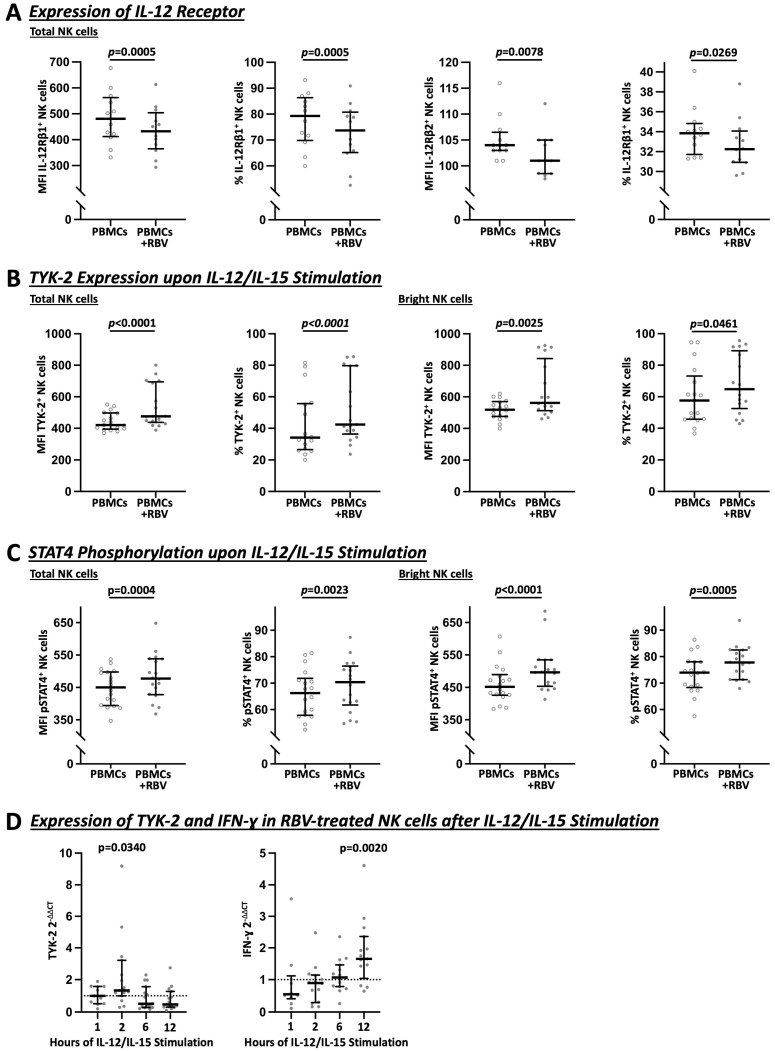
Flow cytometric (**A**–**C**) and PCR (**D**) analysis of IL-12 signaling in total and bright NK cells under RBV treatment. In (**A**–**C**), MFI and frequencies compared. (**A**) Expression of IL-12 receptors on NK cells (n = 12). (**B**) Expression of TYK-2 upon IL-12/IL-15 stimulation (n = 17). (**C**) Expression of pSTAT4 upon IL-12/IL-15 stimulation (n = 18). (**D**) Gene expression of TYK-2 and IFN-γ upon IL-12/IL-15 stimulation (n = 10–15 and n = 18). Appearance: median with interquartile range, statistical analysis: Wilcoxon matched-pairs signed-rank test (**A**–**C**) or one-sample Wilcoxon signed-rank test (**D**).

**Figure 6 cells-12-00453-f006:**
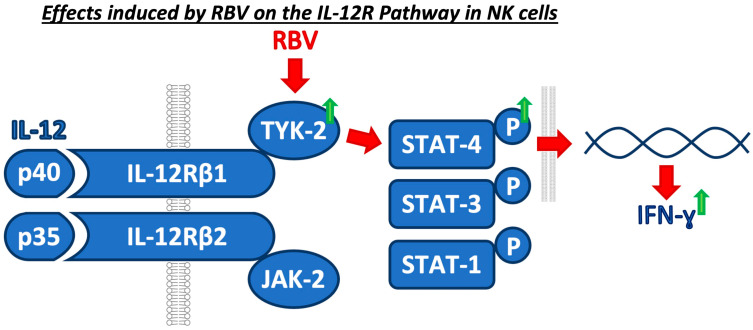
Concluding graphical summary of the identified mechanism of RBV on the IL-12 pathway.

## Data Availability

The original data presented in the study are available upon request from the corresponding author.

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
