# Peer review of "Immunomodulation of Natural Killer Cell Function by Ribavirin Involves TYK-2 Activation and Subsequent Increased IFN-γ Secretion in the Context of In Vitro Hepatitis E Virus Infection"

_cells, 2023, doi:10.3390/cells12030453_

Round 1

Reviewer 1 Report

Kupke et al investigated the immunomodulation of RBV on NK cells in the context of an in-vitro HEV infection and propose a mechanism by which RBV increases IFN-g production by NK cells. The data supporting this mechanism in this article are also weak, more relevant evidence need to be provided. Several issues make interpretations difficult to draw.

Major comments

1.    There are few supporting data for RBV to exert antiviral effects through NK cells, only Figure 1D. More data are needed to support that the antiviral effect of RBV was dependent on NK cell.

2.    The effect of NK cell depletions from PBMCs on HEV replication was investigated through viral titer. The levels of HEV antigen in NK cell depletions and NK cell un depletions should also be analyzed.

3.    More experimental methods, such as Western blotting, Elisa, should be used to verify the results in this study. The results of the effect of RBV treatment on NK cell receptors and a mechanism by which RBV increases IFN-g production by NK cells were obtained using flow cytometry alone.

4.    The antigen of HEV is often considered to be positively correlated with the level of intracellular RNA replication. However, Figure 1A and Figure 1C showed that the antigen titer increased significantly when RNA replication was significantly reduced when the cells were treated with PBMCs alone and RBV and PBMCs together. How do authors explain their contradicting findings. This should be discussed.

5.     The effect on activatory and inhibitory NK cell receptors was divergent. Among these receptors, which were activatory receptors. The receptors should be described detailly. Divergent effect of RBV treatment on NK cell receptors should be discussed.

6.    As manuscript mentioned in 297-299, RBV-treated NK cells showed a slight down-regulation of both IL-12 receptor subunits IL-12Rb1 and IL-12Rb2, whether there is any influence on the subsequent stimulation tests of IL-12 and IL-15.

7.    The up-regulation of related molecules was observed through the stimulation of RBV, and the mediated pathway between them was speculated by the time of the increase. Is there more direct evidence to prove the relationship between them, for example, prove TYK-2 activation upregulate of STAT-4 or inhibit kinases in the pathway with inhibitors to probe the effect on IFN-γ.

8.    Line 274-275, “To investigate the activity and the ability of NK cells to produce IFN-g, the cells were stimulated with IL-12 and IL-15”. Figure 4 showed different IL stimulations, including IL-12, IL-15, and IL-18. Please explain the reason that IL-18 was included in the Figure 4 but not Figure 3.

Minor comments

1.       Please describe the source or production method of the virus.

2.       Description of the MOI detection method needs to be added.

3.       The authors describe that PBMCs were isolated from healthy platelet donors. Were the HEV infection and anti-HEV immune response status in these donors identified. These donors should be tested for HEV-related markers, including HEV RNA, anti-HEV IgM, anti-HEV IgG, HEV antigen.

4.       The cells stimulated with different concentrations of IL-12, IL-15, and IL-18. How were the concentrations of IL-12, IL-15, and IL-18 determined.

5.       The p-values were calculated by one-tailed or two- tailed?

6.       Please describe the full name at first appearance, such as DPBS, c/ml.

7.       2.8 Antibodies for flow cytometry. Were these antibodies used in the NK cell staining for flow cytometry? Describe the methods and the reagents together, not separately.

8.       The method of the depletion of CD19+ cells was not included in the manuscript.

9.       What is the untreated HEV- controls in Figure. 1B. The HEV-Luc+ HepaRG was the untreated HEV- controls? Figure. 1B seem to showed the effect of HEV but not PBMCs on the healthy HepaRG cells. The differences between HEV-Luc+ HepaRG and HEV-Luc+ HepaRG+PBMCs and between HEV+Luc+ HepaRG and HEV+Luc+ HepaRG+PBMCs need to described.

10.    Lines 203-204, Appearance: median with interquartile range (min to max). Figure 1 showed median with range (min to max) or median with interquartile range?

11.    please supplement the p-value between HEV+ HepaRG cells+PBMCs +RBV and HEV+ HepaRG cells+RBV in Fig1A.

Reviewer 2 Report

Summary Statement:

Kupke et al use an in vitro hepatoma model to examine the effect of ribavirin (RBV) on NK cells during infection with a genotype 3 HEV strain; the predominant genotype found in chronically HEV-infected solid organ transplant recipients (SOTRs). SOTRs represent a vulnerable patient population, and treatment options are currently limited to a trial of reduction in immunosuppressive therapy and treatment with RBV monotherapy. A better understanding of the immunomodulatory mechanisms of RBV during HEV infection may provide insights into treatment outcomes in patients.

            The authors choose to focus on the effect of RBV on NK cells, given that these are highly expressed in liver cells during HEV infection relative to other hepatitis viruses. HepaRG cells were infected with HEV subtype 3c and subsequently co-cultured with PBMCs from healthy platelet donors +/- RBV. Analyses were performed at 24h. Their results demonstrate that RBV and NK cells have an additive effect in depleting viral titers, and that RBV upregulates NK cell activation markers while downregulating NK cell cytotoxicity.

            The authors subsequently conduct experiments to show that RBV stimulates IFN-É£ expression when cells are co-stimulated with IL-12 and IL-15. They then seek to map where in the JAK-STAT pathway RBV is acting, and find that IL-12 receptors are downregulated, therefore RBV is presumably acting downstream to the receptor. They find that TYK2 is upregulated and therefore deduce that RBV is acting upon TYK2 to increase IFN-É£ levels.

Reviewer Comments:

            The work is overall interesting, well-written, and reads logically, however some clarification is needed regarding the conclusions from the experiments in Figures 4 and 5:

1)    If RBV is presumably acting via TYK-2, why is co-stimulation with IL-12 and IL-15 required to see any IFN-É£ expression? Why do the authors suppose that no IFN-É£ stimulation is seen when cells are treated with RBV alone (Figures 4A and 4B)?

2)    If IL-12 receptor levels are downregulated, how do the authors suppose that RBV increases sensitivity to IL-12?

Round 2

Reviewer 1 Report

The authors responded the comments accordingly. The article was revised and recommended to be published.